# Genome-wide association analyses identify two susceptibility loci for pachychoroid disease central serous chorioretinopathy

Yoshikatsu Hosoda et al.[#]

The recently emerged pachychoroid concept has changed the understanding of age-related macular degeneration (AMD), which is a major cause of blindness; recent studies attributed AMD in part to pachychoroid disease central serous chorioretinopathy (CSC), suggesting the importance of elucidating the CSC pathogenesis. Our large genome-wide association study followed by validation studies in three independent Japanese and European cohorts, consisting of 1546 CSC samples and 13,029 controls, identified two novel CSC susceptibility loci: *TNFRSF10A-LOC389641* and near *GATA5* (rs13278062, odds ratio = 1.35, $P = 1.26 \times 10^{-13}$; rs6061548, odds ratio = 1.63, $P = 5.36 \times 10^{-15}$). A T allele at *TNFRSF10A-LOC389641* rs13278062, a risk allele for CSC, is known to be a risk allele for AMD. This study not only identified new susceptibility genes for CSC, but also improves the understanding of the pathogenesis of AMD.

---

[#]A full list of authors and their affiliations appears at the end of the paper.

 

Central serous chorioretinopathy (CSC) is a common eye disease characterized by serous retinal detachment of the macular regions and retinal pigment epithelium (RPE)[1,2]. Although retinal detachments in CSC eyes typically resolve spontaneously, some cases become chronic resulting in permanent retinal tissue damage[3–5]. Additionally, recent studies have shown that the occurrence of choroidal neovascularization (CNV) is a common complication, leading to severe vision loss[6,7]. Based on these findings, CSC is currently recognized as an important sight-threatening disease that can lead to legal blindness. Although clinical studies have shown that dysfunction of the retinal pigment epithelium and hyperpermeability of the choroidal vessels are essential in the etiology of CSC[8], the exact mechanisms underlying CSC pathology remain unknown. Previous studies reported that the risk factors of CSC include glucocorticoid use, increased adrenergic hormones, stress, male gender, *Helicobacter pylori* infection, and type A personality[1,5,9–16].

Recently, the pathological overlap between age-related macular degeneration (AMD), which is a major cause of legal blindness in developed countries, and CSC has received increased attention as they share similar clinical characteristics, including serous retinal detachment, pigment epithelium detachment, and CNV occurrence. Particularly, because CNV secondary to CSC (which was recently named as pachychoroid neovasculopathy because of its characteristic thick choroid[17–19]) often masquerades as AMD[20], recent studies highlighted the importance of differentiating AMD and CSC[18,19,21–25]. Interestingly, although AMD and CSC show similar clinical manifestations, genetic studies revealed that they have contrasting characteristics with respect to *complement factor H* (*CFH*), an established AMD susceptibility gene[26]; the risk alleles for AMD at *CFH* Y402H and *CFH* I62V confer a protective effect against CSC[27–29]. Taken together, investigating the genetic background of CSC is currently of great importance and can improve the understanding of the etiology of both CSC and AMD.

Two genome-wide association studies (GWASs) for CSC have been reported[30,31]. However, both studies have limitations such as a lack of replication studies using independent cohorts and limited sample sizes. Even after considering previous candidate gene studies[28,32–34], no CSC susceptibility gene other than *CFH* has been well-replicated and established. Therefore, robust GWASs are needed to further understand the genetic background of CSC.

In the present study, we conducted a large-scale GWAS for CSC followed by replication analyses using three independent Japanese and European cohorts. Our analysis using 1546 CSC cases and 13,029 controls identified two novel CSC susceptible loci, rs13278062 at *TNFRSF10A-LOC389641* and rs6061548 near *GATA5*, which were significantly associated with the disease on a genome-wide scale. As these single-nucleotide polymorphisms (SNPs) showed a robust and homogenous effect among the Japanese and European cohorts, they may be important targets for further molecular biological evaluation and the development of new treatments.

## Results

### GWAS for CSC and replication analyses in Japanese cohorts.

After stringent quality control, 2,893,743 SNPs were included in the first-stage of the GWAS in a Japanese case–control cohort consisting of 610 CSC patients and 2850 controls. We included three principal components as covariates to adjust for possible population stratification, which provided an acceptable control; the genomic inflation factor lambda ($\lambda_{GC}$) was 1.157. The quantile–quantile plot is shown in Supplementary Fig. 1.

To examine whether a population substructure existed and its influence on the GWAS results, we performed principal component analysis (PCA) for the current study using publicly available multiethnic genotype data from 1000 Genome project (Phase 3, ftp://ftp.1000genomes.ebi.ac.uk/vol1/ftp/release/20130502/) (Supplementary Fig. 2) and without this data (Supplementary Fig. 3). The analyses revealed that nearly all subjects in our discovery GWAS fell into the Japanese cluster, whereas a mild population substructure existed within the current discovery GWAS.

In the first-stage, we identified two loci showing a suggestive *P*-value of $<1.0 \times 10^{-6}$ for rs13278062 at *TNFRSF10A-LOC389641* ($OR_{discovery} = 1.38$, $P_{discovery} = 5.94 \times 10^{-7}$) and rs6061548 near *GATA5* ($OR_{discovery} = 1.61$, $P_{discovery} = 2.52 \times 10^{-7}$). The Manhattan plot, regional plots and linkage disequilibrium plots are shown in Figs. 1 and 2 and Supplementary Fig. 4, respectively. Downstream of rs13278062, some SNPs in the *CHMP7* region showed relatively low *P*-values.

No variants were observed downstream of rs6061548 in the regional plot probably owing to stringent QC. Therefore, we made a regional plot around rs6061548 using the GWAS results before QC was applied. In this regional plot, many SNPs within *GATA5* showed low *P*-values; the most significant association was found at rs13044490 within *GATA5* ($OR = 1.67$, $P = 2.94 \times 10^{-10}$, Supplementary Fig. 5).

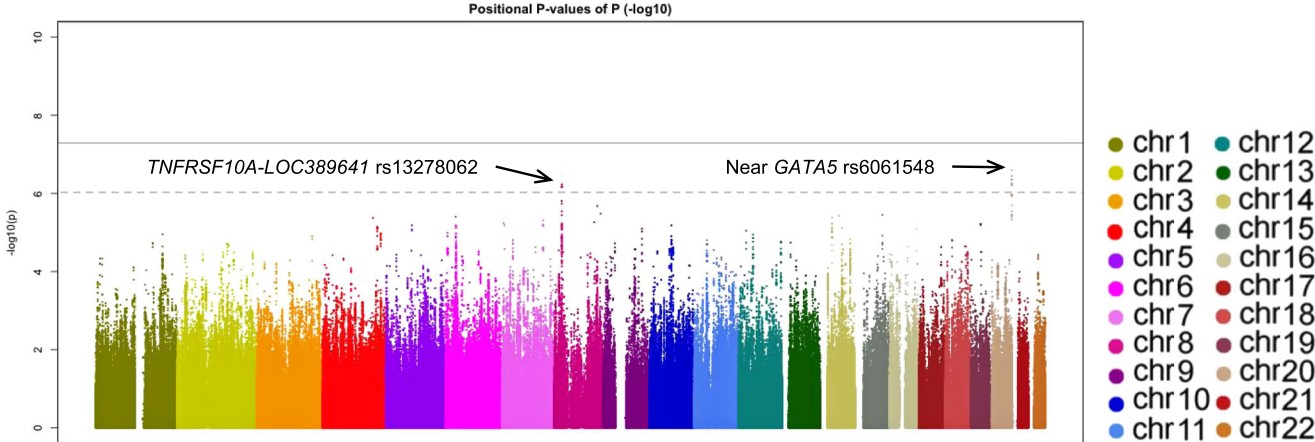

**Fig. 1 Manhattan plot for discovery GWAS using 610 patients with central serous chorioretinopathy and 2580 control participants.** Each plot shows −log10-transformed *P*-values for all SNPs. The horizontal solid line represents the genome-wide significance threshold of $P = 5.0 \times 10^{-8}$, and the lower broken line represents the suggestive threshold of $P = 1.0 \times 10^{-6}$. Two SNPs exceeded the suggestive threshold; rs13278062 at *TNFRSF10A-LOC389641* ($P = 5.94 \times 10^{-7}$) and rs6061548 near *GATA5* ($P = 2.52 \times 10^{-7}$).

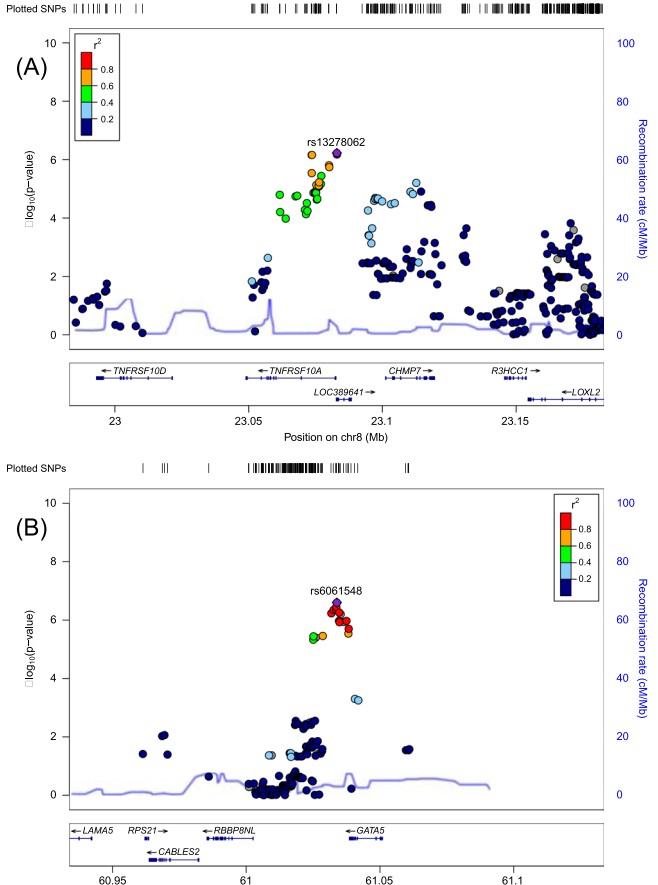

**Fig. 2 Regional association plots of evaluated SNPs around two suggestive SNPs in discovery GWAS.** Plots represent the −log10 (P-values) obtained from the first-stage GWAS. Each plot corresponds to the following; **a** *TNFRSF10A-LOC389641* and **b** near *GATA5* regions.

In the replication stage using 278 independent CSC samples from Japan, both rs13278062 at *TNFRSF10A-LOC389641* (OR = 1.35, $P = 8.97 \times 10^{-4}$) and rs6061548 near *GATA5* (OR = 1.39, $P = 1.28 \times 10^{-2}$) showed significant associations with CSC (Table 1).

The associations of two SNPs with CSC were further evaluated in another Japanese CSC case–control dataset. Using the data from Kobe University Hospital, we found that rs6061548 near *GATA5* was also significantly associated with CSC in this dataset (OR = 2.29, $P = 3.26 \times 10^{-6}$). Rs13278062 showed a trend towards an association with the same direction of effect (OR = 1.19, $P = 0.189$, Table 1).

**Replication in a European cohort and meta-analysis.** We conducted trans-ethnic replication analyses to confirm the association. Association analysis using 521 cases and 3577 controls of European descent revealed that rs13278062 and rs6061548 were significantly associated with CSC in the European cohort (Table 1). The odds ratios of the SNPs in the European cohort were similar to that observed in the Japanese cohort ($OR_{European} = 1.36$, $P_{European} = 1.47 \times 10^{-5}$ for rs13278062, and $OR_{European} = 1.60$, $P_{European} = 5.80 \times 10^{-4}$ for rs6061548). A meta-analysis of the Japanese and European cohorts also revealed strong association between CSC and rs13278062 at *TNFRSF10A-LOC389641* ($OR_{meta-all} = 1.35$, $P_{meta-all} = 1.26 \times 10^{-13}$), as well as rs6061548 near *GATA5* ($OR_{meta-all} = 1.63$, $P_{meta-all} = 5.36 \times 10^{-13}$). As shown in Fig. 3, the

effects of rs13278062 and rs6061548 on CSC occurrence were homogenous among the Japanese and European cohorts.

**Association of previously reported SNPs with CSC.** Previous GWAS of CSC reported two susceptibility loci, *CFH* rs1329428 and *SLC7A5* rs11865049[30,31]. In the current GWAS, *CFH* rs1329428 showed a significant association with CSC (OR = 1.17, $P = 0.015$). *SLC7A5* rs11865049 showed the same direction of effect as previously reported but did not reach a significant value (OR = 1.15, $P = 0.24$). These results are summarized in Table 2.

**Expression in human tissue.** A search in a publicly available quantitative trait locus analysis (eQTL) database revealed that rs13278062 was significantly associated with *TNFRSF10A* expression (GTEx Portal. https://gtexportal.org/home/), but not with any other genes nor *CHMP7*. A multi-tissue eQTL plot revealed that the normalized effect size (NES) of rs13278062 on *TNFRSF10A* expression was strongest in the adrenal gland (NES = −0.973, $P = 4.5 \times 10^{-39}$; Supplementary Fig. 6). Although it was unclear whether rs6061548 near *GATA5* affects the expression or function of any genes in the database, rs13044490 within *GATA5*, which was a top-hit SNP in the regional plot, was significantly associated with *GATA5* expression. The effect of rs13044490 on *GATA5* expression was strongest in the esophageal muscularis (NES = 0.347, $P = 3.9 \times 10^{-8}$; Supplementary Fig. 7).

To confirm the expression of the genes in the human retina and choroid that play a role in CSC pathogenesis, we conducted database searching using the Eyeintegration database (https://eyeintegration.nei.nih.gov/, v1.01) and The Ocular Tissue Database (https://genome.uiowa.edu/otdb), the only databases that includes gene expression data in human retina and choroid. The Eyeintegration database showed that the expression of both *TNFRSF10A* and *GATA5* in the adult human RPE/choroid (n = 48) was stronger than that in the adult human retina (n = 52) as summarized in Fig. 4. These results were supported by The Ocular Tissue Database, which shows that the expression of *TNFRSF10A* in the adult human RPE/choroid was stronger than that in the adult human retina (PLIER normalized expression level = 18.90 vs. 16.34) and that of *GATA5* in the adult human RPE/choroid was also stronger than that in the adult human retina (PLIER normalized expression level = 57.26 vs. 48.30). The Eyeintegration database revealed that the expression of both *TNFRSF10A* and *GATA5* was also observed in other human tissues (Supplementary Note).

**Pathway analysis.** We performed pathway analysis using VEGAS2Pathway (https://vegas2.qimrberghofer.edu.au/). In total, 9723 pathways were evaluated. The top ten pathways are shown in Supplementary Table 1. The most significantly associated pathway was the ESCRT-III complex (M00412, $P = 2.60 \times 10^{-5}$). However, no pathways reached the genome-wide, pathway-based significant P-value of $<1.0 \times 10^{-5}$[30,35].

## Discussion
In the current study, we identified two novel susceptible loci, rs13278062 at *TNFRSF10A-LOC389641* and rs6061548 near *GATA5* for a common pachychoroid disease, CSC, through a large GWAS followed by replication studies in three independent case–control datasets of Japanese and European origin. These SNPs showed robust and consistent association among all datasets. Interestingly, rs13278062 has been reported to be a susceptibility SNP for AMD. As the relationship between AMD and CSC has received increased attention, the current results have

**Table 1 Discovery and replication analyses to identify SNPs associated with CSC.**

| SNP | | rs13278062 | | | | | | rs6061548 | | | | | |
|---|---|---|---|---|---|---|---|---|---|---|---|---|---|
| Nearby genes | | *TNFRSF10A-LOC389641* (in gene) | | | | | | *GATA5* (Nearby) | | | | | |
| Effect allele | | T | | | | | | T | | | | | |
| CHR: position | | 8: 23082971 | | | | | | 20: 61033892 | | | | | |
| | | CSC | | Control | | OR | P | CSC | | Control | | OR | P |
| Stage | Ethnicities | N | EAF | N | EAF | (95% CI) | | N | EAF | N | EAF | (95% CI) | |
| Discovery GWAS | Japanese | 610 | 0.421 | 2850 | 0.345 | 1.38 (1.22–1.57) | $5.94 \times 10^{-7}$* | 610 | 0.138 | 2850 | 0.088 | 1.64 (1.36–1.98) | $2.52 \times 10^{-7}$* |
| Replication stage 1 | Japanese | 277 | 0.392 | 5449 | 0.324 | 1.35 (1.13–1.60) | $8.97 \times 10^{-4}$* | 278 | 0.128 | 4546 | 0.095 | 1.39 (1.07–1.80) | 0.0128* |
| Replication stage 2 | Japanese | 137 | 0.409 | 1153 | 0.368 | 1.19 (0.92–1.53) | 0.189* | 137 | 0.161 | 1153 | 0.077 | 2.29 (1.60–3.27) | $5.55 \times 10^{-6}$* |
| Replication stage 3 | European | 521 | 0.591 | 3577 | 0.520 | 1.36 (1.18–1.56) | $1.47 \times 10^{-5}$† | 521 | 0.082 | 3577 | 0.051 | 1.60 (1.23–2.07) | $5.80 \times 10^{-4}$† |
| Meta-analysis of Japanese Data (discovery + replication 1+ replication 2) | | | | | | | | | | | | | |
| | Japanese | 1024 | | 9452 | | 1.34 (1.22–1.48) | $1.45 \times 10^{-9}$ | 1025 | | 8549 | | 1.64 (1.43–1.89) | $3.38 \times 10^{-12}$ |
| Meta-analysis of Japanese Data and European Data | | | | | | | | | | | | | |
| | Japanese and European | 1545 | | 13,029 | | 1.35 (1.24–1.46) | $1.26 \times 10^{-13}$ | 1546 | | 12,126 | | 1.63 (1.44–1.85) | $5.36 \times 10^{-15}$ |

All meta-analyses were performed using a fixed-effect model
*CHR* chromosome, *EAF* effect allele frequency, *CSC* central serous chorioretinopathy, *OR* odds ratio, *CI* confidence interval
*P-values were derived using logistic regression analysis. †P-values were derived using association analysis

**Table 2 Association of previously reported SNPs on GWAS study with CSC.**

| SNP | Genes | CHR | Position | Effect allele | EAF in CSC (N = 610) | EAF in control (N = 2850) | OR (95% CI) | P* | Effect directions as reported |
|-----|-------|-----|----------|---------------|---------------------|--------------------------|-------------|-----|-------------------------------|
| rs1329428 | CFH | 1 | 196702810 | T | 0.506 | 0.466 | 1.17 (1.03–1.32) | $1.45 \times 10^{-2}$ | + |
| rs11865049 | SLC7A5 | 16 | 87874140 | A | 0.078 | 0.069 | 1.15 (0.91–1.44) | 0.24 | + |

CHR chromosome, EAF effect allele frequency in the discovery stage, CSC central serous chorioretinopathy, OR odds ratio, CI confidence interval
*P-values were derived using logistic regression analysis

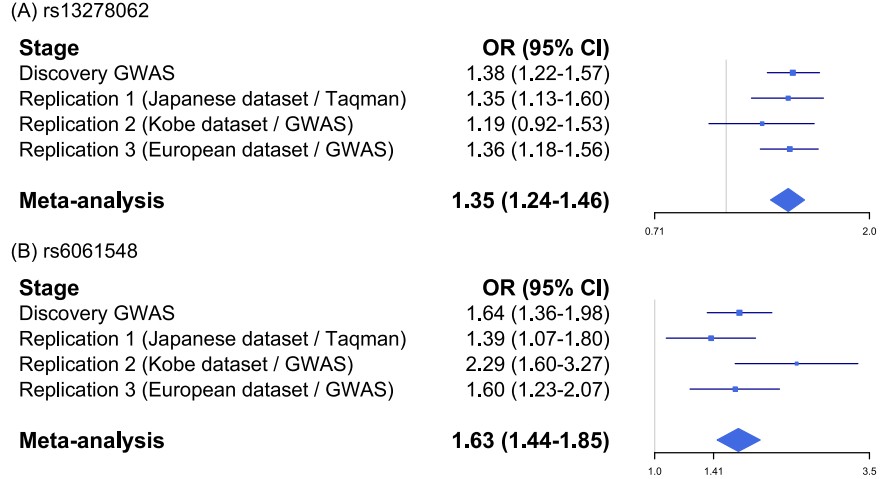

**Fig. 3 Forest plots showing the effects of novel susceptibility SNPs in each cohort and meta-analysis.** Forest plots showing the effects of **a** rs13278062 and **b** rs6061548 on CSC in each cohort and meta-analysis. Both SNPs showed robust, consistent, and mild to moderate association with CSC across ethnic groups.

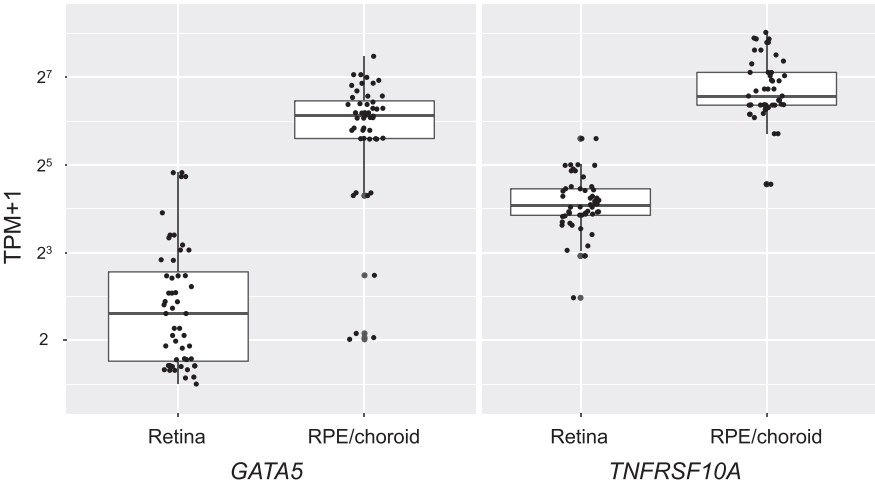

**Fig. 4 Boxplots of _TNFRSF10A_ and _GATA5_ expression levels in the human retina and RPE/choroid.** Expression levels of _TNFRSF10A_ and _GATA5_ in the human retina (n = 52) and RPE/choroid (n = 48) are given in transcripts per million (TPM). _TNFRSF10A_ was strongly expressed in the human RPE/choroid compared to in the retina (116.62 ± 58.53 vs. 18.11 ± 8.96 TPM, P < 0.001). _GATA5_ was also strongly expressed in the human RPE/choroid compared to in the retina (69.05 ± 37.91 vs. 4.66 ± 7.08 TPM, P < 0.001). TPM values are expressed as the mean ± standard deviation and compared by Wilcoxon test. RPE retinal pigment epithelium.

important scientific implications that improve the understanding of the similarities and differences between AMD and CSC.

_TNFRSF10A_ was first identified as an AMD susceptibility locus in a Japanese population[36]. Although this association has been confirmed in other ethnicities, the effect of _TNFRSF10A_ on AMD in Caucasians was reported to be weaker than that in Asian population[37–39]. Recently, some researchers reported a subgroup within AMD that incorporates the characteristics of CSC, such as

thick choroid and choroidal vascular hyperpermeability[7,17,19]. Although the precise rate of the subgroup among patients with AMD is currently unknown, the rate is estimated to be higher in Asians than in Caucasians[22,40]. Taken together with the current result, we speculate that eyes with CNV, which is traditionally diagnosed as AMD, include CNV secondary to CSC, which may occur more frequently in Asians compared to that Caucasians. In support of this, the effect of _TNFRSF10A_ on CSC occurrence

(OR = 1.38) in the present study was higher than that of tradi-tional AMD occurrence in a previous study ($OR_{forAMD}$ = 1.25 in Asian, and $OR_{forAMD}$ = 1.11 in European)[39]. It is also possible that *TNFRSF10A* has pleiotropic effects on both AMD and CSC. The effects of *TNFRSF10A* on AMD should be further evaluated while stratifying the data for the presence of a pachychoroid background.

CNV accompanied by the characteristics of CSC was recently termed as pachychoroid neovasculopathy, as the thickened choroid is the most characteristic clinical feature of the condition. Gene expression evaluation in ocular tissue supports the impor-tance of the RPE/choroid regarding the etiology of pachychoroid neovasculopathy and CSC, as *TNFRSF10A* is more strongly expressed in the RPE/choroid than in the retina. However, the exact role of *TNFRSF10A* in CSC occurrence or choroidal structure is unclear. Considering that the adrenergic hormones are established risk factors for CSC[41,42], and that the eQTL showed that the genotype of rs13278062 at *TNFRSF10A* was strongly associated with its expression in the adrenal gland, *TNFRSF10A* may affect the risk of CSC by modulating hormone secretion from the adrenal glands.

We additionally identified that rs6061548 near *GATA5* was associated with CSC. *GATA5* is known to play an important role in vascular system development[43–45]. A recent study showed that *GATA5* is expressed in the microvascular endothelial cells and that its inactivation in mice results in vascular endothelial dysfunc-tion[46]. Considering the stronger expression of *GATA5* in the RPE/ choroid than in the retina, we speculate that *GATA5* may affects the susceptibility to CSC through vascular endothelial dysfunction in the choriocapillaris, which constitutes the inner vascular layers of the choroid, composed largely of fenestrated capillaries. This hypothesis is compatible with previous reports showing that eyes with CSC had choriocapillary hypoperfusion[47,48], and primary choroidal ischemia[48]. *GATA5* is also known to play an important role in stomach development and gastric diseases[49–51]. Interest-ingly, *GATA5* is reported to be upregulated by *H. pylori* infec-tion[52], which is one of the risk factors of CSC[10,12–14]. As described above, *GATA5* may also be an important target for the further understandings of CSC.

The current study has many strengths and limitations. The main strengths are its large sample size and that multiple replica-tion studies were performed in different ethnic groups, which led to robust results. As CSC is thought to be a benign, self-limiting disease, its genetic background has not been clarified. However, the pathology of CSC has become an important issue in relation to AMD. To our knowledge, this study is the largest genetic study on CSC and the first study to identify transetheni-cally robust susceptible genes using a non-hypothesis-driven, exploratory approach. Considering the identification of relatively high, consistent effects across multiple ethnic groups, the current study has strong scientific implications. However, the sample size of this study was still limited. An even larger sample size is required to identify additional susceptibility SNPs with low allele frequency or low effect size and to further elucidate disease pathways in CSC. Another limitation is the inflated $\lambda_{GC}$ in our discovery GWAS, which may have led to false-positive associa-tions. This inflation may be related to the presence of a mild population substructure, as inflation was still observed even after conducting the GWAS using samples genotyped with a single-DNA microarray platform, OmniExpress ($\lambda_{GC}$ = 1.098). How-ever, the positive associations of both *TNFRSF10A-LOC389641* rs13278062 and near *GATA5* rs6061548 were still significant even after genomic control correction ($P_{meta}$ = $2.57 \times 10^{-12}$ and $P_{meta}$ = $6.29 \times 10^{-12}$, respectively; Supplementary Table 2), and thus these associations appear to be robust.

In summary, we identified two novel CSC susceptibility loci, rs13278062 at *TNFRSF10A-LOC389641* and rs6061548 near *GATA5*, using a total of 1546 CSC cases and 13,029 controls. These variants showed robust, consistent, and mild to moderate associations with CSC across ethnicities. We confirmed that both genes are expressed in the choroid, which is the main focus of CSC and AMD. Our findings improve the understanding of the pathogenesis of CSC and AMD.

## Methods

**Patient enrollment**. In the discovery GWAS, 610 Japanese patients with CSC were recruited from the Kyoto University Hospital, and 2850 healthy Japanese controls were recruited from the Aichi Cancer Center Research Institute, Hayashi Eye Hospital, Mizoguchi Eye Hospital, Oita University Faculty of Medicine, Ideta Eye Hospital, Shinjo Eye Clinic, Miyata Eye Hospital, Ozaki Eye Hospital, Kyoto University Hospital, and Nagahama City Hospital. Detailed information of the control cohort is described elsewhere[53], and is briefly summarized in the Supple-mentary Note and Supplementary Table 3.

In the first replication stage, 278 CSC patients were recruited from across Japan, particularly Kyoto University Hospital ($N$ = 57), Kagawa University Hospital ($N$ = 104), Yamanashi University Hospital ($N$ = 80), and Fukushima Medical University Hospital ($N$ = 37). Control allele frequency data were extracted from the genome-wide dataset of healthy Japanese subjects from the Tohoku Medical Megabank Organization ($N$ = 3498)[54–56], Kyoto University ($N$ = 3074)[57,58], and Yokohama City University dataset ($N$ = 1048). Detailed information is shown in the Supplementary Note. In the second and the third replication stages, the Kobe CSC case–control dataset, which consists of 137 Japanese patients with CSC and 1153 Japanese controls, and European CSC case–control dataset, which consists of 521 European patients with CSC and 3577 European controls, were utilized, respectively. Although the detailed information for this dataset has been described previously[30,31], a brief summary is also provided in the Supplementary Note.

All procedures adhered to the tenets of the Declaration of Helsinki. The Institutional Review Board and Ethics Committee of each participating institute approved the respective study protocols. All patients were fully informed of the purpose and procedures of the study, and written consent was obtained from all patients prior to their participation in the study.

**Diagnosis of patients with CSC**. All patients underwent a comprehensive oph-thalmic examination, including visual acuity measurement; slit-lamp biomicro-scopy; dilated fundoscopy; color fundus photography; and optical coherence tomography (OCT) examination, including enhanced depth imaging, fundus autofluorescence, fluorescein angiography, and indocyanine green angiography CSC was diagnosed based on medical history, serous retinal detachment, thickened choroid with dilated choroidal vessels seen on OCT, choroidal vascular hyper-permeability on ndocyanine green angiography, and/or leakages on fluorescein angiography at the level of the RPE. Based on these findings, two retina specialists (M.M. and Y.H.) diagnosed the patients independently, and discrepancies were adjusted in a face-to-face consensus session. Patients with other causes of fluor-escein leakage or serous retinal detachment unrelated to CSC (e.g., AMD, intrao-cular inflammation, diabetic retinopathy, or retinal vein occlusion) were excluded from the study. In the second and third replication stages, patients with CSC were diagnosed based on previously described criteria[29–31]. The details are summarized in the Supplementary Note.

**Genotyping**. Genomic DNA was extracted from peripheral blood samples according to standard laboratory procedures. A series of BeadChip DNA arrays (Illumina, San Diego, CA, USA), namely OmniExpress ($N$ = 250) and Asian Screening Array ($N$ = 360), were used for genotyping the CSC samples, while Human610-Quad BeatChip ($N$ = 1194) and OmniExpress ($N$ = 1656) were used for genotyping the control samples.

Genotype imputation was performed using the Michigan imputation server (https://imputationserver.sph.umich.edu/index.html#!pages/home) with the 1000 Genome dataset (phase3 v5 release) of East Asians as a reference panel for each dataset. In each dataset, SNPs with a call rate <90% or a minor allele frequency <1% were excluded before genotype imputation. Imputed SNPs for which $R^2$ was <0.9 were excluded from the subsequent association analysis. Next, SNPs with a call rate <90%, a minor allele frequency <1%, or significant deviation ($P < 1.0 \times 10^{-5}$) from Hardy–Weinberg equilibrium were excluded from further statistical analysis, and samples with a call rate <90% were also excluded. We checked the allelic discrimination of SNPs showing a suggestive association with CSC in the discovery GWAS ($P < 1.0 \times 10^{-5}$) for each platform. We excluded SNPs with an insufficient quality of allelic discrimination and their proxy SNPs ($R^2 > 0.8$). Finally, 2,893,743 SNPs from 610 CSC samples and 2850 control samples were used for discovery stage analysis.

In the first replication stage, genotypes of CSC samples ($N$ = 278) were determined using a commercially available assay (TaqMan SNP assay with the ABI

PRISM 7700 system; Applied Biosystems, Foster City, CA, USA). We extracted the allele frequency from existing healthy Japanese genome-wide datasets, which included the Tohoku dataset[54–56], Kyoto University[57,58], and Yokohama City University datasets. The details of the genotyping methods are summarized in the Supplementary Note. Briefly, the Integrative Japanese Genome Variation Database (ver 3.5 K JPN, https://ijgvd.megabank.tohoku.ac.jp/) provides genomic reference panels obtained from 3554 normal Japanese subjects recruited from the Tohoku Medical Megabank Organization, Iwate Medical Megabank Organization, Nagahama Prospective Cohort for Comprehensive Human Bioscience, and National Hospital Organization Nagasaki Medical Center. All DNA samples were whole-genome-sequenced using the Illumina HiSeq 2500. The Human Genetic Variation Database is a database of genomic reference panels released from Kyoto University (http://www.hgvd.genome.med.kyoto-u.ac.jp/index.html). This database is a web-accessible resource of genetic variations in the Japanese population. Whole-genome SNV genotyping was performed for 3712 individuals, who formed a subset of participants of The Nagahama Prospective Genome Cohort for the Comprehensive Human Bioscience (the Nagahama Study), with the Illumina HumanHap610 quad, Omni 2.5 M and Human exome Beadarrays (Illumina). The Yokohama City University dataset includes 1048 Japanese healthy controls recruited from the Yokohama City University, Okada Eye Clinic, and Aoto Eye Clinic in Yokohama, Kanagawa Prefecture, Japan. Genotypes of samples from Yokohama City University were determined using BeadChip DNA arrays, namely Human OmniExpress chip (Illumina), with the standard protocol recommended by each manufacturer.

The Kobe CSC case–control dataset, which was used in the second replication stage, was genotyped using the Human OmniExpress BeadChips (Illumina, San Diego, CA, USA) as described elsewhere[31]. Genomic imputation was performed for the dataset using the BEAGLE 4.1 and 1000 genomes dataset (phase3 v5 release) as the reference panels. Imputed SNPs for which $R^2 < 0.7$ were excluded from the imputed dataset. The association of SNPs with CSC was tested by logistic regression analysis with no adjustment. The European CSC case–control dataset, which was used in the third replication stage, was genotyped using OmniExpress-12 or OmniExpress-24chip. The data were imputed with the Haplotype Reference Consortium release 1.1.2016, and were stringent quality controls were performed as described previously[30].

**Statistical analyses**. Genome-wide logistic regression analysis was conducted for the CSC, adjusting for three principal components. As on age and sex was not available for 1656 out of the 2850 control samples, adjustment for these factors was not performed in the discovery GWAS. Experiment-wide significance in the discovery stage was set to $P = 5.0 \times 10^{-8}$. We carried SNPs with $P$-value of $<1.0 \times 10^{-6}$ forward to the replication stage. In the first and second replication stages, logistic regression was performed for the SNPs that showed suggestive association in the first-stage. All meta-analyses were performed using the fixed-effect model. Thereafter, differences were considered significant at $P < 0.05$. Deviations in genotype distributions from Hardy–Weinberg equilibrium were assessed with chi-square tests. These statistical analyses were performed with R ver 3.5.2 (R Foundation for Statistical Computing, Vienna, Austria; available at http://www.rproject.org/), GCTA ver. 1.25.3 (http://cnsgenomics.com/software/gcta/index.html#Overview) and PLINK ver. 2.0 (https://www.cog-genomics.org/plink/2.0/).

In the third replication stage using the European dataset, association analysis was performed using the Firth bias-corrected likelihood ratio test, implemented in EPACTS (version 3.2.6, https://genome.sph.umich.edu/wiki/EPACTS; University of Michigan), correcting for sex and the first two components of ancestry analysis.

**Pathway analysis**. Using the GWAS summary statistics and $P$-values of replicated SNPs, we performed gene-based analysis and clustered genes into pathways using VEGAS2pathway (https://vegas2.qimrberghofer.edu.au/, version 2). Briefly, VEGAS2 successively prunes the list of variants with $R^2$ criteria of 0.99, 0.90, 0.70, and 0.50, if a gene contains >1000 SNPs. After each pruning interval, VEGAS2 checks the number of pruned SNPs. If the number of pruned SNPs is <1000, VEGAS2 uses the pruned SNPs from that interval to perform gene-based analysis; otherwise, it iteratively applies an increasingly stringent $R^2$ criteria on all SNPs in the gene. After applying a pruning criterion of $R^2 = 0.50$, it uses all pruned SNPs for analysis irrespective of the number. The gene list was obtained from the VEGAS2 official site (https://vegas2.qimrberghofer.edu.au/glist-hg19). Obtained gene-based result was used to perform calculate pathway-based tests and empirical $P$-values to obtain the significance of each pathway. Regions +/− 0 kb outside of genes were defined as gene regions, and all SNPs were used for analysis. For the SNPs that were carried forward for to the replication stage, $P$-values from meta-analysis were used rather than GWAS $P$-values. The Biosystems gene-pathway annotation file was obtained from the VEGAS2 official site (https://vegas2.qimrberghofer.edu.au/biosystems20160324.vegas2pathSYM). The significance threshold of the empirical $P$-value in the pathway analysis was set to $1.0 \times 10^{-5}$[30,35].

**Database search**. To determine the existence of a population substructure and its influence on the GWAS results, publicly available genotype data from five populations, including African, South Asian, European, East Asian, and Japanese

(1000 Genome project, Phase 3, ftp://ftp.1000genomes.ebi.ac.uk/vol1/ftp/release/20130502/) were used for PCA.

Searching of the publicly available quantitative trait locus analysis (eQTL) database (Genotype-Tissue Expression (GTEx) Portal: https://gtexportal.org/home/) revealed an association between the SNP genotypes and gene expression in multiple human tissues. NES is defined as the slope of the linear regression and is computed as the effect of the alternative allele relative to the reference allele.

Expression of genes in the adult human retina and RPE was explored in the eyeintegration database (https://eyeintegration.nei.nih.gov/, v1.01. accessed 10 April 2019). The database lists the expression levels of genes given in transcripts per million. Gene expression levels were compared by Wilcoxon test. Expression of genes in the human retina and choroid was explored in The OCULAR TISSUE DATABASE (https://genome.uiowa.edu/otdb/, accessed 28 February 2019). This database provides quantitative expression level of genes in ocular tissues determined by the PLIER (Probe Logarithmic Intensity Error) algorithm.

**Reporting summary**. Further information on research design is available in the Nature Research Reporting Summary linked to this article.

**Data availability**
Top SNPs ($n = 100$) in the discovery GWAS can be available as Supplementary Data 1. The source data about the expression of genes in the adult human retina and RPE can be available as Supplementary Data 2. The complete GWAS summary data can be visualized here: https://figshare.com/articles/CSC_control_PC3_assoc_logistic/11136047. The datasets generated during the current study are also available from the corresponding author on reasonable request.

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

## Acknowledgements

We thank Ms. Hatsue Hamanaka for assistance with genotyping. We thank for Dr. Masato Akiyama for supervising about method of the pathway analysis.

## Author contributions

Y.H., M.M., R.Y., F.M., K.Y., and A.T. designed the study; Y.H., M.M., R.L.S., C.J.F.B., C.B.H., A.Mi., A.Me., Y.S., S.Y., Y.T., M.H., Y.M., H.N., A.O., S.O., H.T., A.U., M.M., A.Tak., N.U.A., A.Taj., T.S., N.M., C.S., T.I., S.H., E.K.D.J., A.I.D.H., K.Y., and A.T. performed the study; Y.H., M.M., R.L.S., C.J.F.B., C.B.H., A.Mi., A.Me., A.Tak., T.S., N.M., C.C.K., T.Y.W., R.Y., S.H., E.K.D.J., A.I.D.H., K.Y., and A.Taj. analyzed the data; and Y.H., M.M., and K.Y. prepared the manuscript.

## Competing interests

The authors declare no competing interests.

## Additional information

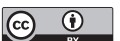

Yoshikatsu Hosoda[1,2], Masahiro Miyake[1,2]*, Rosa L. Schellevis[3], Camiel J.F. Boon[4], Carel B. Hoyng[3], Akiko Miki[5], Akira Meguro[6], Yoichi Sakurada[7], Seigo Yoneyama[7], Yukari Takasago[8], Masayuki Hata[1], Yuki Muraoka[1], Hideo Nakanishi[1], Akio Oishi [1], Sotaro Ooto[1], Hiroshi Tamura [1], Akihito Uji[1], Manabu Miyata [1], Ayako Takahashi[1], Naoko Ueda-Arakawa[1], Atsushi Tajima [9], Takehiro Sato [9], Nobuhisa Mizuki[6], Chieko Shiragami [8], Tomohiro Iida[10], Chiea Chuen Khor [11,12], Tien Yin Wong[11,13,14], Ryo Yamada[2], Shigeru Honda[5,15], Eiko K. de Jong [3], Anneke I. den Hollander [3], Fumihiko Matsuda[2], Kenji Yamashiro [1,16,17] & Akitaka Tsujikawa[1,17]

[1]Department of Ophthalmology and Visual Sciences, Kyoto University Graduate School of Medicine, Kyoto, Japan. [2]Center for Genomic Medicine, Kyoto University Graduate School of Medicine, Kyoto, Japan. [3]Department of Ophthalmology, Donders Institute of Brain, Cognition and Behaviour, Radboud University Medical Centre, Nijmegen, The Netherlands. [4]Department of Ophthalmology, Leiden University Medical Center, Leiden, The Netherlands. [5]Department of Surgery, Division of Ophthalmology, Kobe University Graduate School of Medicine, Kobe, Japan. [6]Department of Ophthalmology and Visual Sciences, Yokohama City University School of Medicine, Yokohama, Japan. [7]Department of Ophthalmology, University of Yamanashi, Faculty of Medicine, Yamanashi, Japan. [8]Department of Ophthalmology, Kagawa University Faculty of Medicine, Kagawa, Japan. [9]Department of Bioinformatics and Genomics, Graduate School of Advanced Preventive Medical Sciences, Kanazawa University, Kanazawa, Ishikawa, Japan. [10]Department of Ophthalmology, Tokyo Women's Medical University, Tokyo, Japan. [11]Singapore National Eye Centre, Singapore Eye Research Institute, Singapore, Singapore. [12]Division of Human Genetics, Genome Institute of Singapore, Singapore, Singapore. [13]Saw Swee Hock School of Public Health, National University of Singapore and National University Health System, Singapore, Singapore. [14]Ophthalmology and Visual Sciences Academic Clinical Program, Duke-NUS Medical School, National University of Singapore, Singapore, Singapore. [15]Department of Ophthalmology and Visual Sciences, Osaka City University Graduate School of Medicine, Osaka, Japan. [16]Department of Ophthalmology, Otsu Red-Cross Hospital, Otsu, Japan. [17]These authors contributed equally: Kenji Yamashiro, Akitaka Tsujikawa. *email: miyakem@kuhp.kyoto-u.ac.jp

