## [Peer review file · Communications Biology]

Reviewers' comments:

Reviewer #1 (Remarks to the Author):

In this genome-wide association study (GWAS), the authors discovered two single nucleotide polymorphisms (SNPs), rs13278062 at the TNFRSF10A-LOC389641 locus and rs6061548 near GATA5 to be associated with central serous chorioretinopathy (CSC). The significant associations were replicated in two independent Japanese cohorts and a European population. This study added new information to our knowledge about the genetic architecture of CSC. Some issues in this study should be clarified before being considered for publication.

Major issues:

1. As the discovery cohort was recruited across Japan, adding a scatter plot of the first three principal components would be very informative for the readers to understand population substructures.
2. Was the entire reference panel of 1000 Genome data used in the imputation? Or any of the sub-populations?
3. It was not clear how the association was done in the Kobe dataset. Logistic regressions were also adjusted for three principal components? Moreover, the authors should be advised to adjust for age and gender in the regression analysis as they were known factors altering the risk of CSC.
4. The lambda and QQ plot indicated the existence of potential systematic errors. Did the authors explore the sources of potential errors? What would explain the inflated lambda?
5. The SNP rs13044490 in GATA5 was not selected for replication in other cohorts. The authors may want to remove the comment - "We additionally identified that rs6061548 and rs13044490 near/at GATA5 were associated with CSC." Or the authors could replicate rs13044490 in all the cohorts.
6. There are some other genes in the TNFRSF10A-LOC389641 locus. The authors might want to discuss that. Are the expression of TNFRSF10A and GATA5 specific in the eye tissues?
7. The pathway analysis would be based on validated/replicated GWAS datasets. Some of the positive SNPs in the discovery cohort might not be true significant findings.

Minor points

1. The names of the cohorts in Figure 3 were confusing. The Kobe dataset was a GWAS, which should serve as replication #1? The "Caucasian dataset" was the European cohort.

Reviewer #2 (Remarks to the Author):

In this manuscript, the investigators performed a GWAS study using a large sample size and multiple replication studies in different ethnic groups. They found two new potential risk factors of central serous chorioretinopathy (CSC). Since there have been several GWAS study in CSC, this research is an important addition in Ophthalmology field. However it may not acquire much attention in general research community unless there are functional study.

Reviewer #3 (Remarks to the Author):

Comments to the authors:

"Genome-wide association analyses identify two novel susceptibility loci for pachychoroid disease central serous chorioretinopathy"

This manuscript presents results from the largest genome-wide association study of CSC to date. Using Japanese discovery dataset and replication from both Japanese and European datasets, the authors were able to identify two novel susceptibility loci for CSC.

I have couple of major comments for the authors of this manuscript:

1. The cases and controls of the discovery dataset and the Japanese replication dataset were genotyped using different genotyping chip. As the possible differences in genotype calling may introduce allele frequency differences between the datasets, these may appear as false positive

findings when comparing the cases and controls. This could also explain the fairly high genomic inflation factor in the association results. Have the authors thought of the possible implications of their data design?

2. Was the imputation of the datasets performed separately for each of the different genotyping dataset or were the datasets first combined and then imputed? Imputation of cases and controls separately may introduce similar bias as above.

3. What were the QC steps performed before genotype imputation? How were the datasets combined before imputation (if they were combined)? Was the whole 1000G imputation reference used or only the matching population subset?

4. Regarding the third locus that achieved genome-wide significant association; for me this association peak is as plausible as the other two and disregarding a locus from the follow-up analysis based on "some imputation errors" when the imputation quality threshold has been already set high (0.9) seems reverse cherry-picking. Are all the three associated SNPs in this locus imputed in every dataset you have, or were some of them genotyped? Why is the SNP coverage in the results in that area so sparse?

5. In the discussion section the authors write about the differences in the SNP effects between Japanese and Europeans. Could this difference be due to the possible bias explained in comments 1-2 above which is not present in the European replication set where the cases and controls were genotyped together?

6. Why weren't the discovery association analyses adjusted by age and sex which are usually included as covariates in every association study to date?

In addition to these methodological comments, the manuscript would benefit from thorough spell check.

Referee expertise:

Referee #1: Ocular genomics

Referee #2: GWAS of eye disease

Referee #3: GWAS

Reviewer 1

Reviewer's comment

1. As the discovery cohort was recruited across Japan, adding a scatter plot of the first three principal components would be very informative for the readers to understand population substructures.

Response to Reviewer

Thank you for your helpful advice. As the reviewer suggested, presenting the possible population substructure would be quite informative for the readers. According to the reviewer's comment, we have provided two figures;

(1) a scatter plot of principal components using the current discovery GWAS and data from 1000 genomes project (JPT, AFR, EAS, EUR, and SAS) and

(2) a scatter plot of principal components within Japanese samples in the current study.

(1)

(2)

Plot (1) indicates that samples in the current discovery GWAS (orange plots) belong to the known Japanese cluster (JPT, blue plots), indicating no significant population structure from the macro perspective.

In contrast, from the relatively micro perspective, some population substructure is observed within the current discovery GWAS, as shown in plot (2). This mild population substructure may be one reason for the genomic inflation in the current study. Please refer to our reply to reviewer's comment 4.

Changes in the Manuscript

- We included plots (1) and (2) as Supplementary Figure 2 and Supplementary Figure 3, respectively.
- We described these results in the Result sections as follows;
“We included three principal components as covariates to adjust for possible population stratification, which provided an acceptable control; the genomic inflation factor lambda (λ_{GC}) was 1.157. The quantile-quantile plot is shown in Supplementary Figure 1.
To examine whether a population substructure existed and its influence on the GWAS results, we performed principal component analysis (PCA) for the current study using publicly available multiethnic genotype data from 1000 Genome project (Phase 3, <ftp://ftp.1000genomes.ebi.ac.uk/vol1/ftp/release/20130502/>) (Supplementary Figure 2) and without this data (Supplementary Figure 3). The analyses revealed that nearly all subjects in our discovery GWAS fell into the Japanese cluster, whereas a mild population substructure existed within the current discovery GWAS.”

(page 7, line 113-119)
- Regarding the population substructure, we have discussed the inflated λ_{GC} in accordance with reviewer's comment 4 and the other reviewer's comment. Please refer to the response to comment 4.

Reviewer's comment
2. Was the entire reference panel of 1000 Genome data used in the imputation? Or any of the sub-populations?
Response to Reviewer
Thank you for your comment. Although we mentioned this point in the Supplementary Note, it was difficult for the readers to find, as the reviewer pointed out. We have moved this information from the Supplementary Note to the Methods section.
Changes in the Manuscript
“Genotype imputation was performed using the Michigan imputation server (https://imputationserver.sph.umich.edu/index.html#!pages/home) with the 1000 Genome dataset (phase3 v5 release) of East Asians as a reference panel for each dataset. In each dataset, SNPs with a call rate <90% or a minor allele frequency <1% were excluded before genotype imputation.” (page 15, line 324)

Reviewer's comment
3-1. It was not clear how the association was done in the Kobe dataset. Logistic regressions were also adjusted for three principal components?
Response to Reviewer
Thank you for your comment. In the Kobe dataset, the association was tested by logistic regression analysis without any adjustment. As this information was not included in the manuscript, we have clarified this point in the Methods section.
Changes in the Manuscript
“Genomic imputation was performed for the dataset using the BEAGLE 4.1 and 1000 genomes dataset (phase3 v5 release) as the reference panels. Imputed SNPs for which $R^2 < 0.7$ were excluded from the imputed dataset. The association of SNPs with CSC was tested by logistic regression analysis with no adjustment.” (page 16, line 347-348)

Reviewer's comment					
3-2. Moreover, the authors should be advised to adjust for age and gender in the regression analysis as they were known factors altering the risk of CSC.					
Response to Reviewer					
Thank you for this helpful advice. Unfortunately, information on age and sex was not available for some of the control samples (1,656/2,850), and thus we could not adjust for these factors in the discovery GWAS. However, according to the reviewer's advice, we performed logistic regression analysis adjusting for age, sex, and three principal components using samples for which age and sex were available. The results for 2 SNPs (rs13278062 and rs6061548) are summarized below.					
SNP	Gene	This study (adjusting for 3 principal components)		Logistic regression analysis adjusting for age, sex, and three principal components	
		N	OR	N	OR
rs13278062	TNFRSF10A-LOC389641	3,460	1.38 (1.22–1.57)	1,804	1.39 (1.19–1.62)
rs6061548	GATA5		1.64 (1.36–1.98)		1.77 (1.39–2.26)
As shown, the effect sizes of these SNPs after adjusting for age and sex (right column) were similar to those without adjustment (left column). Thus, the current result was not biased by this limitation, at least for these two SNPs.					
Although we agree with the reviewer that adjusting for age and sex would improve the efficiency of our discovery GWAS, we did not make this change in the revised manuscript because for the reasons described above.					
Changes in the Manuscript					
-					

Reviewer's comment
4. The lambda and QQ plot indicated the existence of potential systematic errors. Did the authors explore the sources of potential errors? What would explain the inflated lambda?
Response to Reviewer
Thank you for this important comment. To get straight to the point, the main reason for the inflated lambda value may be the population substructure. We evaluated our data sets for three potential factors to inflate lambda (1) imputation error, (2) the use of different genotyping platforms between cases and controls, and (3) population substructure. (1) imputation error To eliminate the influence of imputation, we conducted a GWAS using only directly genotyped SNPs common among all platforms used. The same QC as our original discovery GWAS was applied. However, lambda was still inflated in this analysis ($\lambda_{GC} = 1.142$). Based on this finding, we considered there were other causes of inflation and imputation error's contribution relatively negligible. (2) the use of different genotyping platforms between cases and controls To eliminate the effect of using different genotyping platforms, we conducted a GWAS using 250 CSC samples and 1656 controls genotyped on the same platform, Omni Express, after adjusting for three principal components. The same QC as our original discovery GWAS was applied. As a result, the lambda was inflated before (1.098) and after imputation (1.107), although the inflation was improved compared to that of our original discovery GWAS (1.157). These results suggest that another reason (i.e., population substructure) was the main cause of the inflation, although the use of different genotyping platforms inflated the lambda to some extent. (3) population substructure As we described in the reply to reviewer's comment 1, there was a mild population substructure, although it was not significant from the macro perspective. Considering the results of (1) and (2), the main reason for the inflated lambda may be the population

substructure.

Because this population substructure could not be eliminated by adjusting for principal components, we performed the most conservative correction for inflation: genomic control correction. However, meta-analysis using METAL software revealed a genome-wide significant association for both *TNFRSF10A* rs13278062 ($P_{GC-corrected} = 2.57 \times 10^{-12}$) and near *GATA5* rs6061548 ($P_{GC-corrected} = 6.29 \times 10^{-12}$), suggesting that the associations of rs13278062 and rs6061548 with CSC occurrence are robust.

As this reviewer's comment is very important, we have discussed this issue in the main body of our manuscript.

Changes in the Manuscript

“An even larger sample size is required to identify additional susceptibility SNPs with low allele frequency or low effect sizes and to further elucidate disease pathways in CSC. **Another limitation is the inflated λ_{GC} in our discovery GWAS, which may have led to false-positive associations. This inflation may be related to the presence of a mild population substructure, as inflation was still observed even after conducting the GWAS using samples genotyped with a single DNA microarray platform, OmniExpress ($\lambda_{GC} = 1.098$, data not shown). However, the positive associations of both *TNFRSF10A-LOC389641* rs13278062 and near *GATA5* rs6061548 were still significant even after genomic control correction ($P_{meta} = 2.57 \times 10^{-12}$ and $P_{meta} = 6.29 \times 10^{-12}$, respectively; data not shown), and thus these associations appear to be robust.”**

(Page 12-13, line 255-262)

Reviewer's comment
5. The SNP rs13044490 in GATA5 was not selected for replication in other cohorts. The authors may want to remove the comment - "We additionally identified that rs6061548 and rs13044490 near/at GATA5 were associated with CSC." Or the authors could replicate rs13044490 in all the cohorts.
Response to Reviewer
Thank you for this advice. According to the reviewer's suggestion, we avoided referring to rs13044490 here.
Changes in the Manuscript
We have revised the manuscript from " We additionally identified that rs6061548 and rs13044490 near/at GATA5 were associated with CSC." to "We additionally identified that rs6061548 near GATA5 was associated with CSC.." (page 11, line 231)

Reviewer's comment

6-1. There are some other genes in the *TNFRSF10A-LOC389641* locus. The authors might want to discuss that.

Response to Reviewer

Thank you for your advice. As the reviewer pointed out, there are a few other genes in the *TNFRSF10A-LOC389641* locus. However, rs13278062 appears to be associated with *TNFRSF10A* for the following reasons:

(1) We found that the top SNP rs13278062 was significantly associated only with *TNFRSF10A* expression in the Gtex portal database (<https://www.gtexportal.org/home/snp/rs13278062>), but not with any other genes.

(2) We also analyzed the linkage disequilibrium pattern around rs13278062, as shown below (Figure). The color scheme shows R-squared values. We found that *TNFRSF10A-LOC389641* was the only gene in the LD block of rs13278062.

As this was not discussed in the original manuscript, we have provided this figure as Supplementary Figure 4 and revised the manuscript text.

Changes in the Manuscript

We have included the figure as Supplementary Figure 4 and revised the manuscript as follows:

“In the first stage, we identified two loci **showing** a suggestive P -value of $<1.0 \times 10^{-6}$ for rs13278062 at *TNFRSF10A-LOC389641* ($OR_{\text{discovery}} = 1.38$, $P_{\text{discovery}} = 5.94 \times 10^{-7}$) and rs6061548 near *GATA5* ($OR_{\text{discovery}} = 1.61$, $P_{\text{discovery}} = 2.52 \times 10^{-7}$). **The Manhattan plot, regional plots, and linkage disequilibrium plots are shown in Figure 1, Figure 2, and Supplementary Figure 4, respectively. Downstream of rs13278062, some SNPs in the *CHMP7* region showed relatively low P values.**”

(Page 7, line 121-126).

“A search in a publicly available quantitative trait locus analysis (eQTL) database revealed that rs13278062 was significantly associated with *TNFRSF10A* expression (GTEx Portal. <https://gtexportal.org/home/>), **not with any other genes nor *CHMP7*.**”

(page 9, line 163-165)

Reviewer's comment

6-2. Are the expression of *TNFRSF10A* and *GATA5* specific in the eye tissues?

Response to Reviewer

Thank you for the comment. As shown below, a publicly available database revealed the expression of both genes in other tissues (eyeintegration, <https://eyeintegration.nei.nih.gov/>).

Because the main focus of CSC is the eye, particularly the retina and choroid tissues in adults, we evaluated the expression of candidate genes in these tissues. As we detected a significant difference in the expression of these genes between the human retina and RPE/choroid, we have revised the figure 4, providing P values for the differences in expression levels.

For the readers to easily access these data, we have provided a link to this figure as a Supplementary Note and revised the manuscript.

Changes in the Manuscript

We have provided a link to this Figure as Supplementary Information and revised our manuscript and Figure 4 as follows:

“The Eyeintegration database showed that the expression of both *TNFRSF10A* and *GATA5* in the adult human RPE/choroid (n = 48) was stronger than that in the adult human retina (n = 52), as summarized in Figure 4. These results were supported by The Ocular Tissue Database, which shows that the expression of *TNFRSF10A* in the adult human RPE/choroid was stronger than that in the adult human retina (PLIER normalized expression level = 18.90 vs 16.34), and that of *GATA5* in the adult human RPE/choroid was also stronger than that in the adult human retina (PLIER normalized expression level = 57.26 vs 48.30). The Eyeintegration database revealed that the expression of both *TNFRSF10A* and *GATA5* was also observed in other human tissues (Supplementary Note).” (page 9, line 178-186)

“Figure 4. Boxplots of *TNFRSF10A* and *GATA5* expression levels in the human retina and RPE/choroid.

Expression levels of *TNFRSF10A* and *GATA5* in the human retina (n = 52) and RPE/choroid (n = 48) are given in transcripts per million (TPM). *TNFRSF10A* was strongly expressed in the human RPE/choroid compared to in the retina (116.62 ± 58.53 vs 18.11 ± 8.96 TPM, $P < 0.001$). *GATA5* was also strongly expressed in the human RPE/choroid compared to in the retina (69.05 ± 37.91 vs 4.66 ± 7.08 TPM, $P < 0.001$). TPM values are expressed as the mean \pm standard deviation and compared by Wilcoxon test.

RPE; retinal pigment epithelium”

Reviewer's comment

7. The pathway analysis would be based on validated/replicated GWAS datasets. Some of the positive SNPs in the discovery cohort might not be true significant findings.

Response to Reviewer

Thank you for your comment. We agree with the reviewer that some of the positive SNPs in the discovery cohort were not true-positives. Particularly, using the very low P-values for SNPs in *CYP2A7* during the discovery stage, which were not replicated in subsequent analysis, was inappropriate.

In response to the reviewer's comment, we have made two changes:

(1) addition of a QC step

Through the revision process, we found that rs10411264 near *CYP2A7* showed poor allelic discrimination (refer to the figure on the right). Thus, we performed a manual allelic discrimination check as an additional QC step.

Because it is not possible to check allelic discrimination for millions of SNPs, we evaluated allelic discrimination for SNPs showing suggestive ($P < 1.0 \times 10^{-5}$) associations with CSC in the discovery GWAS. SNPs exhibiting poor allelic discrimination and their proxy SNPs ($R^2 > 0.8$) were excluded during this QC step.

As a result, several SNPs, including rs10411264 near *CYP2A7*, were excluded, and we revised the results of discovery GWAS (Figure 1 and Figure 2).

(2) using P values derived from meta-analysis

We used P values derived from the meta-analysis for rs13278062 at *TNFRSF10A-LOC389641* and rs6061548 near *GATA5*, rather than those from the discovery GWAS.

In case our understanding of the application of VEGAS2Pathway was not correct, we reviewed the concept and usage of VEGAS2Pathway as summarized below.

VEGAS, VEGAS2, and VEGAS2Pathway are designed to account for sub-threshold SNPs to improve the power of analysis. Although the details of VEGAS2Pathway analysis are provided on web

<https://www.cambridge.org/core/journals/twin-research-and-human-genetics/article/novel-approach-for-pathway-analysis-of-gwas-data-highlights-role-of-bmp-signaling-and-muscle-cell-differentiation-in-colorectal-cancer-susceptibility/2B99124878BB41F67EBE15C6DA97FBF3>), we briefly explain the program:

VEGAS2Pathway is a two-step pathway analysis strategy.

First, VEGAS2 performs a gene-based test. For each gene definition, *P*-values for all SNPs within the gene are first converted to upper tail χ^2 statistics with one degree of freedom (df) and then summed to calculate a gene-based test statistic, accounting for linkage disequilibrium (LD) and gene size (number of SNPs). This approach enabled us to test for the enrichment of multiple SNPs associated with the disease that individually have too modest effects on the disease to reach genome-wide significance using a per-SNP test.

Second, for each set of pre-specified gene-sets, the relevant gene-based results are carried forward in a pathway-based test. To give users a pathway analysis platform useful for a wide range of complex traits, VEGAS2 provides a biosystems gene-pathway annotation file. In the pathway-based test, the relevant gene-based *P* values were first converted to upper-tail χ^2 statistics with one degree of freedom before summing. The summed statistics are used for pathway-based test statistics.

Thus, in VEGAS2Pathway analysis, the inputs are not necessarily based on validated/replicated GWAS datasets.

We also revised the text in the Results and Methods sections related to VEGAS2 Pathway analysis using our revised dataset.

Changes in the Manuscript

Results section

“We performed pathway analysis using VEGAS2Pathway (<https://vegas2.qimrberghofer.edu.au/>). In total, 9,723 pathways were evaluated. The top 10 pathways are shown in Supplementary Table 2. The most significantly associated pathway was the ESCRT-III complex (M00412, $P = 2.60 \times 10^{-5}$). However, no pathways reached the genome-wide, pathway-based significant P-value of less than 1.0×10^{-5} .^{30,35}”

(Page10, line190-191)

Methods section

“The gene list was obtained from the VEGAS2 official site (<https://vegas2.qimrberghofer.edu.au/glist-hg19>). The obtained gene-based result was used to perform pathway-based tests and empirical P-values to obtain the significance of each pathway. Regions +/-0 kb outside of genes were defined as gene regions, and all SNPs were used for analysis. For SNPs that were carried forward to the replication stage, P values from meta-analysis were used rather than GWAS P values. The biosystems gene-pathway annotation file was obtained from the VEGAS2 official site (<https://vegas2.qimrberghofer.edu.au/biosystems20160324.vegas2pathSYM>). The significance threshold of the empirical P value in the pathway analysis was set to 1.0×10^{-5} .^{30,35}”

(Page 17-18, line381-389)

We revised Figure 1, Figure2 and Table 1 as follows;

“Figure 1. Manhattan plot for discovery GWAS using 610 patients with central serous chorioretinopathy and 2,580 control participants.

Each plot shows $-\log_{10}$ -transformed P -values for all SNPs. The horizontal solid line represents the genome-wide significance threshold of $P = 5.0 \times 10^{-8}$, and the lower broken line represents the suggestive threshold of $P = 1.0 \times 10^{-6}$. Two SNPs exceeded the suggestive threshold; rs13278062 at *TNFRSF10A-LOC389641* ($P = 5.94 \times 10^{-7}$) and rs6061548 near *GATA5* ($P = 2.52 \times 10^{-7}$).”

“Figure 2. Regional association plots of evaluated SNPs around two suggestive SNPs in discovery GWAS. Plots represent the $-\log_{10}$ (P -values) obtained from the first-stage GWAS. Each plot corresponds to the following; (A) *TNFRSF10A-LOC389641* and (B) near *GATA5* regions.”

Reviewer's comment																																																													
Minor points																																																													
1. The names of the cohorts in Figure 3 were confusing. The Kobe dataset was a GWAS, which should serve as replication #1? The "Caucasian dataset" was the European cohort.																																																													
Response to Reviewer																																																													
Thank you for your advice. We have revised the manuscript accordingly.																																																													
Changes in the Manuscript																																																													
“Figure 3: Forest plots showing the effects of (A) rs13278062 and (B) rs6061548 on CSC in each cohort and meta-analysis Both SNPs showed robust, consistent, and mild to moderate associations with CSC across ethnic groups.”																																																													
  (A) rs13278062     Stage N (CSC) N (Control) OR (95% CI)   Discovery GWAS 610 2,850 1.38 (1.22-1.57)   Replication 1 (Japanese dataset / Taqman) 277 5,449 1.35 (1.13-1.60)   Replication 2 (Kobe dataset / GWAS) 137 1,153 1.19 (0.92-1.53)   Replication 3 (European dataset / GWAS) 521 3,577 1.36 (1.18-1.56)   Meta-analysis 1,545 13,029 1.35 (1.24-1.46)   (B) rs6061548     Stage N (CSC) N (Control) OR (95% CI)   Discovery GWAS 610 2,850 1.64 (1.36-1.98)   Replication 1 (Japanese dataset / Taqman) 278 4,546 1.39 (1.07-1.80)   Replication 2 (Kobe dataset / GWAS) 137 1,153 2.29 (1.60-3.27)   Replication 3 (European dataset / GWAS) 521 3,577 1.60 (1.23-2.07)   Meta-analysis 1,546 12,126 1.63 (1.44-1.85)  				(A) rs13278062					Stage	N (CSC)	N (Control)	OR (95% CI)	Discovery GWAS	610	2,850	1.38 (1.22-1.57)	Replication 1 (Japanese dataset / Taqman)	277	5,449	1.35 (1.13-1.60)	Replication 2 (Kobe dataset / GWAS)	137	1,153	1.19 (0.92-1.53)	Replication 3 (European dataset / GWAS)	521	3,577	1.36 (1.18-1.56)	Meta-analysis	1,545	13,029	1.35 (1.24-1.46)	(B) rs6061548					Stage	N (CSC)	N (Control)	OR (95% CI)	Discovery GWAS	610	2,850	1.64 (1.36-1.98)	Replication 1 (Japanese dataset / Taqman)	278	4,546	1.39 (1.07-1.80)	Replication 2 (Kobe dataset / GWAS)	137	1,153	2.29 (1.60-3.27)	Replication 3 (European dataset / GWAS)	521	3,577	1.60 (1.23-2.07)	Meta-analysis	1,546	12,126	1.63 (1.44-1.85)
(A) rs13278062																																																													
Stage	N (CSC)	N (Control)	OR (95% CI)																																																										
Discovery GWAS	610	2,850	1.38 (1.22-1.57)																																																										
Replication 1 (Japanese dataset / Taqman)	277	5,449	1.35 (1.13-1.60)																																																										
Replication 2 (Kobe dataset / GWAS)	137	1,153	1.19 (0.92-1.53)																																																										
Replication 3 (European dataset / GWAS)	521	3,577	1.36 (1.18-1.56)																																																										
Meta-analysis	1,545	13,029	1.35 (1.24-1.46)																																																										
(B) rs6061548																																																													
Stage	N (CSC)	N (Control)	OR (95% CI)																																																										
Discovery GWAS	610	2,850	1.64 (1.36-1.98)																																																										
Replication 1 (Japanese dataset / Taqman)	278	4,546	1.39 (1.07-1.80)																																																										
Replication 2 (Kobe dataset / GWAS)	137	1,153	2.29 (1.60-3.27)																																																										
Replication 3 (European dataset / GWAS)	521	3,577	1.60 (1.23-2.07)																																																										
Meta-analysis	1,546	12,126	1.63 (1.44-1.85)																																																										

Reviewer 2

Reviewer's comment

In this manuscript, the investigators performed a GWAS study using a large sample size and multiple replication studies in different ethnic groups. They found two new potential risk factors of central serous chorioretinopathy (CSC). Since there have been several GWAS study in CSC, this research is an important addition in Ophthalmology field. However it may not acquire much attention in general research community unless there are functional study.

Response to Reviewer

Thank you for your comments. We agree with the reviewer that functional studies would provide information useful to the general research community. However, this study is an important addition to the ophthalmology field, as you mentioned. We are planning to perform functional studies including animal experiments in our further analyses.

Changes in the Manuscript

-

Reviewer 3

Reviewer's comment

1. The cases and controls of the discovery dataset and the Japanese replication dataset were genotyped using different genotyping chip. As the possible differences in genotype calling may introduce allele frequency differences between the datasets, these may appear as false positive findings when comparing the cases and controls. This could also explain the fairly high genomic inflation factor in the association results. Have the authors thought of the possible implications of their data design?
2. Was the imputation of the datasets performed separately for each of the different genotyping dataset or were the datasets first combined and then imputed? Imputation of cases and controls separately may introduce similar bias as above.

Response to Reviewer

Thank you for your helpful comments.

As you commented, it is possible that using different genotyping platforms resulted in a high genomic inflation factor.

To eliminate the effect of using different genotyping platforms, we performed a GWAS using 250 CSC samples and 1656 controls genotyped with the same platform, Omni Express, adjusting for three principal components. The same QC as our original discovery GWAS was applied. As a result, the lambda was still inflated before (1.098) and after imputation (1.107), although inflation was improved compared to that in our original discovery GWAS (1.157). These results suggest that inflation occurred mainly for other reasons (i.e., population substructure), although the use of different genotyping platform inflated the lambda to some extent.

As described in our response to reviewer 1's comment, there was a mild population substructure, although it was not significant from the macro perspective. The main reason for the inflated lambda is likely the population substructure.

Because this population substructure could not be eliminated by adjusting for principal components, we performed the most conservative correction for inflation: genomic control

correction. However, meta-analysis using METAL software revealed the genome-wide significance for the association of both *TNFRSF10A* rs13278062 ($P = 2.57 \times 10^{-12}$) and near *GATA5* rs6061548 ($P = 6.29 \times 10^{-12}$), suggesting that the associations of rs13278062 and rs6061548 with CSC occurrence are robust.

As this reviewer's comment is very important, we have discussed this issue in the main body of our manuscript.

Changes in the Manuscript

“An even larger sample size is required to identify additional susceptibility SNPs with low allele frequency or low effect sizes and to further elucidate disease pathways in CSC. **Another limitation is the inflated λ_{GC} in our discovery GWAS, which may have led to false-positive associations. This inflation may be related to the presence of a mild population substructure, as inflation was still observed even after conducting the GWAS using samples genotyped with a single DNA microarray platform, OmniExpress ($\lambda_{GC} = 1.098$, data not shown). However, the positive associations of both *TNFRSF10A-LOC389641* rs13278062 and near *GATA5* rs6061548 were still significant even after genomic control (GC) correction ($P_{meta} = 2.57 \times 10^{-12}$ and $P_{meta} = 6.29 \times 10^{-12}$, respectively; data not shown), and thus these associations appear to be robust.”**

(Page 12-13, line 255-262)

Reviewer's comment
3. What were the QC steps performed before genotype imputation? How were the datasets combined before imputation (if they were combined)? Was the whole 1000G imputation reference used or only the matching population subset?
Response to Reviewer
Thank you for your advice. Although we mention this information in the Supplementary Note, it was difficult for the readers to find, as the reviewer pointed out. We excluded SNPs with a call rate <90% or minor allele frequency (MAF) < 1% before imputation. We have moved this information from the Supplementary Note to the Methods section.
Changes in the Manuscript
“Genotype imputation was performed using the Michigan imputation server (https://imputationserver.sph.umich.edu/index.html#!pages/home) with the 1000 Genome dataset (phase3 v5 release) of East Asians as a reference panel for each dataset. In each dataset, SNPs with a call rate <90% or a minor allele frequency <1% were excluded before genotype imputation. Imputed SNPs for which R^2 was less than 0.9 were excluded from subsequent association analysis.” (page 15, line 322-327)

Reviewer's comment
4-1. Regarding the third locus that achieved genome-wide significant association; for me this association peak is as plausible as the other two and disregarding a locus from the follow-up analysis based on “some imputation errors” when the imputation quality threshold has been already set high (0.9) seems reverse cherry-picking. Are all the three associated SNPs in this locus imputed in every dataset you have, or were some of them genotyped?
Response to Reviewer
Thank you for your advice. We carefully checked the reason for this peak and found that rs10411264 showed poor allelic discrimination, although it passed our original QC. This poor allelic discrimination introduced biased allele frequency for the SNP, leading to a biased allele frequency and low P-value of proxy SNPs. The details of the four SNPs in this peak including imputed/genotyped status and allelic discrimination are summarized at the bottom of this reply. To address this issue, we added a QC step: manual allelic discrimination check. Because we cannot evaluate allelic discrimination for millions of SNPs, we checked allelic discrimination for SNPs showing a suggestive association ($P < 1.0 \times 10^{-5}$) with CSC in the discovery GWAS. SNPs with poor allelic discrimination and their proxy SNPs ($R^2 > 0.8$) were excluded in this new QC step. As a result, only rs10411264 showed poor allelic discrimination. Thus, the SNP and its proxy SNPs were excluded from the study and the peak disappeared. This new quality control step based on the reviewer's comment improved our study. We appreciate the reviewer's suggestion.

【Tables】 Detailed information for the four SNPs in this peak

Around rs5007415, 4 SNPs initially reached a genome-wide significance level of association.

Imputation information for these SNPs for each chip is summarized below.

rs4105141 (19:41393600), OR = 0.66 (0.58–0.75), P = 9.05×10^{-11} , effect allele = A					
Group	N	EAf	R ²	Imputed/Genotyped	Chip
Case (CSC)	360	0.289	0.908	Imputed	Asian Screening Array
Case (CSC)	250	0.444	0.993	Imputed	Omni Express
Control	1,194	0.452	0.994	Imputed	Human610-Quad
Control	1,656	0.438	0.995	Imputed	Omni Express

rs5007415 (chr19:41393760), OR = 0.65 (0.57–0.74), P = 3.57×10^{-11} , effect allele = A					
Group	N	EAf	R ²	Imputed/Genotyped	Chip
Case (CSC)	360	0.284	0.918	Imputed	Asian Screening Array
Case (CSC)	250	0.438	-	Genotyped	Omni Express
Control	1,194	0.437	-	Genotyped	Human610-Quad
Control	1,656	0.449	-	Genotyped	Omni Express

rs10411264 (chr19:41394336), OR = 0.65 (0.57–0.74), P = 3.57×10^{-11} , effect allele = T					
Group	N	EAf	R ²	Imputed/Genotyped	Chip
Case (CSC)	360	0.278	-	Genotyped	Asian Screening Array
Case (CSC)	250	0.438	0.997	Imputed	Omni Express
Control	1,194	0.449	0.997	Imputed	Human610-Quad
Control	1,656	0.437	0.996	Imputed	Omni Express

rs28472879 (chr19:41395755), OR = 0.65 (0.57–0.74), P = 3.57×10^{-11} , effect allele = A					
Group	N	EAf	R ²	Imputed / Genotyped	Chip
Case (CSC)	360	0.283	0.917	Imputed	Asian Screening Array
Case (CSC)	250	0.437	0.993	Imputed	Omni Express
Control	1,194	0.448	0.993	Imputed	Human610-Quad
Control	1,656	0.437	0.992	Imputed	Omni Express

【Figures】 Allelic discrimination for directly genotyped SNPs in this peak for each platform (in-house data)

As shown in Figure C, the Asian Screening Array was not suitable for genotyping rs10411264, as it led to poor allelic discrimination. In contrast, rs5007415 was effectively genotyped.

A. Human610-Quad BeatChip, rs5007415

B. Omni Express, rs5007415

C. Asian Screening Array, rs10411264

We have described the quality control method in the Methods section. We also revised the results of discovery GWAS (Figure 1 and Figure 2).

Changes in the Manuscript

“Next, SNPs with a call rate <90%, a minor allele frequency <1%, or significant deviation ($P < 1.0 \times 10^{-5}$) from Hardy-Weinberg equilibrium were excluded from further statistical analysis, and samples with a call rate <90% were also excluded. We checked the allelic discrimination of SNPs showing a suggestive association with CSC in the discovery GWAS ($P < 1.0 \times 10^{-5}$) for each platform. We excluded SNPs with an insufficient quality of allelic discrimination and their proxy SNPs ($R^2 > 0.8$). Finally, 2,893,743 SNPs from 610 CSC samples and 2,850 control samples were used for discovery stage analysis.”

(page 15, line 327-334)

“Figure 1. Manhattan plot for discovery GWAS using 610 patients with central serous chorioretinopathy and 2,580 control participants.

Each plot shows $-\log_{10}$ -transformed P -values for all SNPs. The horizontal solid line represents the genome-wide significance threshold of $P = 5.0 \times 10^{-8}$, and the lower broken line represents the suggestive threshold of $P = 1.0 \times 10^{-6}$. **Two** SNPs exceeded the suggestive threshold; rs13278062 at *TNFRSF10A-LOC389641* ($P = 5.94 \times 10^{-7}$) and rs6061548 near *GATA5* ($P = 2.52 \times 10^{-7}$).”

“Figure 2. Regional association plots of evaluated SNPs around two suggestive SNPs in discovery GWAS. Plots represent the $-\log_{10}$ (P -values) obtained from the first-stage GWAS. Each plot corresponds to the following; (A) *TNFRSF10A-LOC389641* and (B) near *GATA5* regions.”

Reviewer's comment

4-2. Why is the SNP coverage in the results in that area so sparse?

Response to Reviewer

Thank you for the comment. Though this peak disappeared in our revised manuscript following our new QC step as explained in the response to above comment, we evaluated the cause of sparse SNP coverage in this area.

We found that most SNPs in this area were excluded because of a low minor allele frequency (MAF < 0.01) and low imputation quality value ($R^2 < 0.9$). Before quality control, there were many SNPs around this region as shown below. Please note again that this peak is biased by allelic discrimination error and disappeared following additional quality control.

Changes in the Manuscript

-

Reviewer's comment
5. In the discussion section the authors write about the differences in the SNP effects between Japanese and Europeans. Could this difference be due to the possible bias explained in comments 1-2 above which is not present in the European replication set where the cases and controls were genotyped together?
Response to Reviewer
Thank you for your advice. It seems that you are referring to the second paragraph of the Discussion section, which describes the role of TNFRSF10A. We apologize for our complicated presentation. Here, we are only discussing previous reports describing the effect of TNFRSF10A on AMD, not the current study. To enable the readers to easily understand whether each sentence refers to the current study or previous study, we have revised the paragraph.
Changes in the Manuscript
“TNFRSF10A was first identified as an AMD susceptibility locus in a Japanese population.³⁶ Although this association has been confirmed in other ethnicities, the effect of TNFRSF10A on AMD in Caucasians was reported to be weaker than that in Asian populations.³⁷⁻³⁹ Recently, some researchers reported a subgroup within AMD that incorporates the characteristics of CSC, such as thick choroid and choroidal vascular hyperpermeability.^{7,17,19} Although the precise rate of the subgroup among patients with AMD is currently unknown, the rate is estimated to be higher in Asians than in Caucasians.^{22,40} Taken together with the current results, we speculate that eyes with CNV, which is traditionally diagnosed as AMD, include CNV secondary to CSC, which may occur more frequently in Asians compared to in Caucasians. In support of this, the effect of TNFRSF10A on CSC occurrence (OR = 1.38) in the present study was higher than that of traditional AMD occurrence in a previous study (OR_{forAMD} = 1.25 in Asians, and OR_{forAMD} = 1.11 in Europeans).³⁹ It is also possible that TNFRSF10A has pleiotropic effects on both AMD and CSC. The effects of TNFRSF10A on AMD should be further evaluated while stratifying the data for the presence of a pachychoroid background.” (page 10-11, line 204-218)

Reviewer's comment					
6. Why weren't the discovery association analyses adjusted by age and sex which are usually included as covariates in every association study to date?					
Response to Reviewer					
Thank you for this advice. Unfortunately, information on age and sex was not available for all control samples (1,656/2,850), and thus we could not adjust for these factors in the discovery GWAS. However, according to the reviewer's advice, we performed logistic regression analysis adjusting for age, sex, and three principal components using samples for which age and sex data were available. The results of 2 SNPs (rs13278062 and rs6061548) are summarized below.					
SNP	Gene	This study (adjusting for 3 principal components)		Logistic regression analysis adjusting for age, sex and three principal components	
		N	OR	N	OR
rs13278062	TNFRSF10A-LOC389641	3,460	1.38 (1.22–1.57)	1,804	1.39 (1.19–1.62)
rs6061548	GATA5		1.64 (1.36–1.98)		1.77 (1.39–2.26)
As shown, the effect sizes of these SNPs after adjusting for age and sex (right column) were similar to those without these adjustments (left column). Thus, the current result is not biased by this limitation, at least for these two SNPs.					
Although we agree with the reviewer that adjusting for age and sex would improve the efficiency of our discovery GWAS, we did not make this change in the revised manuscript for the reasons described above.					
Changes in the Manuscript					
-					

Reviewer's comment
In addition to these methodological comments, the manuscript would benefit from thorough spell check.
Response to Reviewer
Thank for your comment. Our revised manuscript has been subjected to English proofreading again.
Changes in the Manuscript
Please refer to the manuscript with tracked changes.

REVIEWERS' COMMENTS:

Reviewer #1 (Remarks to the Author):

Thanks for the revision. The authors addressed my questions/comments.

Reviewer #3 (Remarks to the Author):

I think that the manuscript has improved notably and that the authors have answered sufficiently to all concerns raised by the reviewers.

As the issue of the used regression models not including age and sex is very important regarding the visibility of the limitations of the study and will raise concern among the readers of the article once published, I would request the authors to write this limitation either to the methods section, explaining why adjustments were not applied or to the limitations sections in the discussion.